PERSPECTIVE

# Enduring questions in regenerative biology and the search for answers

Ashley W. Seifert [1✉], Elizabeth M. Duncan [1✉] & Ricardo M. Zayas [2✉]

The potential for basic research to uncover the inner workings of regenerative processes and produce meaningful medical therapies has inspired scientists, clinicians, and patients for hundreds of years. Decades of studies using a handful of highly regenerative model organisms have significantly advanced our knowledge of key cell types and molecular pathways involved in regeneration. However, many questions remain about how regenerative processes unfold in regeneration-competent species, how they are curtailed in non-regenerative organisms, and how they might be induced (or restored) in humans. Recent technological advances in genomics, molecular biology, computer science, bioengineering, and stem cell research hold promise to collectively provide new experimental evidence for how different organisms accomplish the process of regeneration. In theory, this new evidence should inform the design of new clinical approaches for regenerative medicine. A deeper understanding of how tissues and organs regenerate will also undoubtedly impact many adjacent scientific fields. To best apply and adapt these new technologies in ways that break longstanding barriers and answer critical questions about regeneration, we must combine the deep knowledge of developmental and evolutionary biologists with the hard-earned expertise of scientists in mechanistic and technical fields. To this end, this perspective is based on conversations from a workshop we organized at the Banbury Center, during which a diverse cross-section of the regeneration research community and experts in various technologies discussed enduring questions in regenerative biology. Here, we share the questions this group identified as significant and unanswered, i.e., known unknowns. We also describe the obstacles limiting our progress in answering these questions and how expanding the number and diversity of organisms used in regeneration research is essential for deepening our understanding of regenerative capacity. Finally, we propose that investigating these problems collaboratively across a diverse network of researchers has the potential to advance our field and produce unexpected insights into important questions in related areas of biology and medicine.

Modern scientific research generally reflects Thomas Kuhn's notion of *normal science*, whereby technological advances enable researchers to pursue incremental confirmations of existing theory[1]. This approach does, on occasion, produce unexpected insight into outstanding problems, but its fidelity to conventional frameworks discourages the sort of creative experimentation and custom tool-building that produces paradigm-shifting results. Moreover, deploying new technology just because it is available can create the illusion of advancement, i.e., posing questions that have already been answered with less-advanced methods

[1] Department of Biology, University of Kentucky, Lexington, KY 40506, USA. [2] Department of Biology, San Diego State University, San Diego, CA 92182, USA. ✉email: awseifert@uky.edu; Elizabeth.Duncan@uky.edu; rzayas@sdsu.edu

but are proffered as unresolved in order to apply the latest sophisticated (and usually expensive) technology. This is often done with good intentions, e.g., the hope of discovering something new with more sensitive detection or detail, but ultimately does not actually remove the most critical barriers that must be surmounted to move a field forward[2]. Furthermore, this kind of "modern science" incentivizes specialization and, in doing so, focuses effort away from bigger questions that require interdisciplinary efforts and have the potential to advance multiple fields.

In an attempt to think beyond "*normal science*," we wrote this perspective piece to synthesize and share ideas discussed at a Banbury Center workshop on enduring questions in regenerative biology (Box 1). As a group broadly representative of the regeneration field, we reflected on the significant progress made in the past four decades and discussed the types of ambitious community-led projects that we need to pursue to uncover answers to new and enduring questions. There was a strong feeling that by doing this collectively and with the input of experts from other disciplines, regeneration researchers would be better prepared to harness existing or new technologies and design experiments that can begin to address these questions. When scientific collectives assemble to examine major problems, it motivates collaborative efforts across research groups and disciplines. Knowledge creation in one field often spurs progress in related areas, generating benefits for science far beyond the original goals.

Despite the varied expertise among workshop participants and wide-ranging discussions about how regeneration occurs among diverse species, we found broad agreement on identifying several enduring, fundamental questions where scientists should direct their efforts. Importantly, our task was to identify common problems that are not overly reductionist or specific to a particular organism. In essence, we focused on the forest to identify major driving questions while at the same time considering why some trees remain undescribed, hidden, or unknown. We found common ground on the notion that regeneration remains vastly understudied, in part because the most commonly used model organisms do not have robust regenerative capacities. Unsurprisingly, there was no strong impetus for creating new technologies to specifically study regeneration; the enduring technological hurdle in our field is the application of specific transgenic tools to highly regenerative research models. Thus, the perspective put forth here represents a synthesis of focused discussions that aim to stimulate an exchange of ideas and future collaborations between experts in many fields, including regeneration.

### Which key processes comprise regeneration?

Despite general agreement for defining **regeneration** (see Box 2), what remains ill-defined is the set of component processes that comprise regeneration from induction to resolution. While it is clear that regeneration is induced by significant tissue loss or wounding, when and how regeneration-specific processes can be distinguished from those that occur during wound healing and fibrotic repair remain unresolved. Historically, many researchers studied tissue repair mechanisms holistically across diverse species despite varied healing outcomes, i.e., fibrotic or regenerative[3]. However, modern (1980s-present) wound/tissue repair research has largely been siloed from **epimorphic** regeneration research because studies on wound repair rely heavily on mice, rats, and humans (i.e., non-regenerative species in which wound healing normally generates scar tissue). Similarly, research into epimorphic regeneration has relied on a few highly regenerative invertebrate and vertebrate models (e.g., *Hydra*, planarians, zebrafish, salamanders). As regeneration research expands its scope to include new research organisms that can be genetically manipulated and maintained in a laboratory setting[4] or that enable comparisons of regenerative success and failure between closely related and divergent species, our definition of "regenerative capacity" needs re-evaluation. For instance, does a species' "regenerative capacity" signify the inheritance of a single, albeit complex, trait or a combination of separate, interwoven processes? If the latter, are all component processes required for successful regeneration, or might some tissues/species omit one or more? Conversely, do all non-regenerative organisms diverge at the same stage of this progression? Ultimately, refining the definition of regenerative capacity returns us to a fundamental question: what are the component events that comprise regeneration? Defining these components allows one to determine the extent to which they share similarities with or are distinct from processes that occur during fibrotic repair. Thus, we identified a set of fundamental processes that occur across most regenerative species (summarized in Table 1).

After defining component processes common to regeneration, we considered how these events might vary across an ever-broadening set of organisms. For instance, complex tissue regeneration may be evolutionarily constrained such that the entire regenerative response, including all component processes, is canalized with low variation between species for any specific process or set of interrelated processes (Fig. 1a, b). This could be true even if regenerative ability has evolved multiple times in different lineages. Conversely, regeneration could have arisen via convergent evolution (i.e., homoplasy) and variability among component processes may be large across the entire process set or for most of the processes (Fig. 1c). Alternatively, variation across species could be relatively large for some processes yet low for others, i.e., high conservation of specific processes (Fig. 1d). Defining the processes outlined in Table 1 as component parts of regeneration provides a framework to study each one across organisms and facilitates the generation of testable hypotheses to ask what is lacking (or modified) in non-regenerative organisms. For example, one testable hypothesis is that events one through five outlined in Table 1 constitute a general wound response that

---

**Box 1 | Banbury Center Workshop**

The ideas presented in this paper emerged from discussions at a Cold Spring Harbor Laboratory Banbury Center workshop organized by the authors (for information about The Banbury Center, see https://www.cshl.edu/banbury/). The workshop (Enduring Questions in Regenerative Biology) convened biologists and technologists to consider how basic research can advance our understanding of what endows some species and tissues with regenerative capacity, discuss if new tools and technologies are needed to sustain progress in this endeavor, and to consider how these findings could create the next generation of regenerative therapeutics. The workshop participants worked together to identify enduring questions in regenerative biology and why they persist before exploring collaborative approaches to answer them. The three authors were joined at the Banbury Center workshop by Carrie Adler, Maria Barna, Jeff Biernaskie, Sarah Calve, Joshua Currie, Celina Juliano, Je Hyuk Lee, Malcolm Maden, Francesca Mariani, Phillip Newmark, Bret Pearson, Tania Rozario, Tatiana Sandoval Guzmán, Jennifer Simkin, Mansi Srivastava, Kryn Stankunas, and Bo Wang. We note that in addition to discussions at the Banbury Center, participants contributed to reviewing and editing this paper.

**Table 1 A proposed series of biological processes that together comprise regenerative healing from injury through functional tissue replacement.**

| | Biological Process | Description | General process |
|---|---|---|---|
| 1. | Injury signal/wound sensing | Loss of local cell and tissue integrity (including mechanical properties), extracellular DNA, RNA, Damage Associated Molecular Patterns, Pathogen Associated Molecular Patterns released/produced | Wound healing |
| 2. | Immediate defense responses | Cellular and humoral immunity, reactive oxygen species (ROS) production | Wound healing |
| 3. | Peripheral barrier response/restoration[a] | Epithelial cell migration, planar cell polarity (PCP)/cell re-arrangement | Wound healing |
| 4. | Cell activation/cell cycle re-entry[b] | Cell cycle re-entry of quiescent stem or progenitor cells and differentiated cells | Wound healing |
| 5. | Cell migration to the damage site | Accumulation of tissue-building cells or cells acting as modulators of the regenerative environment | Wound healing |
| 6. | Acquisition of a development-like cell state | Transition to a cell state that permits access to genetic programs typically deployed during embryonic development | Blastema formation |
| 7. | Epimorphosis | Expansion of proliferative and progenitor cell populations | Blastema formation |
| 8. | Morphogenesis | Cell fate specification and pattern formation in a regenerative field and integration with existing tissue | Morphogenesis |
| 9. | Remodeling and scaling/morphallaxis[c] | Growth period, which may be associated with morphogenesis or may be unlinked to morphogenesis (context-dependent) | Morphogenesis/Growth |

[a]May not occur in certain internal tissues.
[b]The term de-differentiation refers to terminally differentiated cells becoming less specialized and acquiring the ability to re-enter the cell cycle (see Box 4). Here, we refer to the general activation/cell cycle re-entry of cells that will participate in regeneration.
[c]After an initial period associated with morphogenesis, additional growth may occur that could be separate from the regenerative process.
This consensus set of basic processes is predicted to occur in every organism under study. The temporal ordering and overlap of these processes were simplified to facilitate cross-species comparison.

occurs in all animals, regardless of subsequent steps, with regeneration requiring a distinct mechanism that transitions tissues into regenerative healing (steps six through nine, Table 1). Another hypothesis is that specific events occur during the early wounding response to trigger regenerative healing.

In line with the second hypothesis, one enduring question is whether a particular injury signal predicts the final healing outcome. For example, is there a unique trigger for wound healing versus regeneration[5]? Many investigators have advanced the hypothesis that particular immune cells and their products are specifically required for regeneration, independent of their role in regulating and resolving inflammation[6]. For instance, studies in several adult regeneration models suggest blocking immune cell infiltration (e.g., monocytes and macrophages) or depleting specific immune cell subtypes (e.g., macrophages) prevents normal wound healing and the transition to regenerative healing[7–11]. In contrast, removing similar immune cell types when trying to stimulate regeneration can enhance the response (microglia)[12]. However, it remains an open question if immune phenotypes exist that regulate and promote regeneration-specific processes. The hypothesis that regeneration-promoting immune cell states exist could be tested by comparing immune system responses across different types of injuries in the same species, i.e., where one injury induces regeneration and the other does not (e.g., lizard tails vs. limbs). This hypothesis could also be tested by comparing the immune response in different aged animals or closely related species in which identical tissues heal via regeneration and fibrotic repair, respectively. Comparing the regenerative response in divergent regenerative species may also offer insight into whether components of the immune response are permissive or instructive relative to regeneration.

Component processes and the transitions between them are also critical to define because they establish a framework for generating specific datasets (i.e., cell type and time-point specific) that can capture cell state changes to compare across species. Specifically, changes in chromatin state or genomic architecture, which impact gene regulatory networks (GRNs) and their outputs, may be broadly conserved at these cellular and temporal transitions[13]. Importantly, cell state changes accompanying reactivation of developmental genes can provide signposts for

exploring chromatin states associated with activation or repression of gene expression and thus provide insight into both activation and constraint of regenerative ability[14]. These types of comparisons also lay the groundwork for evaluating cell-type evolution models as they relate to the presence or absence of regenerative capacity[15,16]. Lastly, breaking regeneration into component processes provides a framework for comparing cellular transitions as they occur during regeneration and embryonic development (Boxes 3 and 4).

## What constitutes the beginning of regeneration?

Cells detect and respond to injury regardless of the healing outcome (regeneration or scarring), which raises another outstanding question: to what degree do different healing trajectories overlap (Fig. 2a–d)? For example, the ERK/MAPK signaling pathway is rapidly induced upon tissue injury in both non-regenerating and regenerating species and inhibition of its activation impairs wound healing and regeneration[5,17–21]. What remains unclear is whether the requirement of ERK/MAPK signaling in regeneration directly results from its role in wound healing or if this pleiotropic pathway induces multiple downstream mechanisms that are separately required for multiple regeneration steps (Table 1). Similarly, are there inductive molecules with the dual capacity to promote regeneration and antagonize fibrosis?[22] In broader terms, is it possible to identify a set of conserved cellular and molecular mechanisms that initiate regeneration and thus define the "beginning" of regeneration? As discussed above, it remains unclear whether the early events of wound healing are common across most species and contexts with the unique regenerative response initiated later, or if the different healing trajectories (regeneration vs. scarring) are established during the healing process (Fig. 2c, d). If the latter, how can we discover molecular signals and cell states specific to regeneration?

To distinguish between regenerative responses and more general repair processes, it can be useful to identify events that occur during both regeneration and embryonic development. As embryogenesis unfolds in a temporal sequence beginning from fertilization, tissues, and organs arise at precise positions and

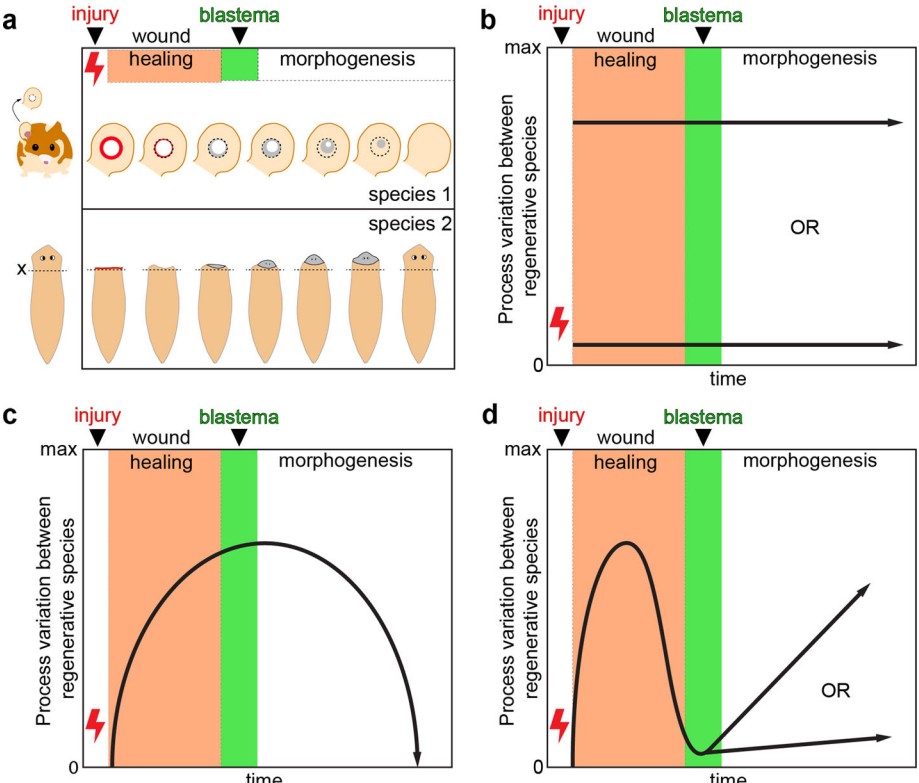

**Fig. 1 Alternative hypotheses explaining potential variability among consensus processes across regenerative organisms. a** Two example species that exhibit epimorphic regeneration: *Acomys cahirinus* (spiny mouse) and *Schmidtea mediterranea* (freshwater planarian). Complex tissue regeneration of the ear pinna (spiny mouse) and head (planarian) is depicted as occurring from an initial tissue injury through complete regeneration. Individual processes as presented in Table 1 are contained within three general phases of regeneration: wound healing, blastema formation, and morphogenesis. Because the timescale over which these processes occur in individual species is highly variable, regeneration in **a** is depicted independent of time. **b**–**d** Three alternative hypotheses describing variability between comparable processes across regenerative species as a function of time. **b** Variability between component processes during the time course of regeneration is either low (i.e., genetic, molecular, and cellular mechanisms are highly conserved across species, bottom arrow) or high (mechanisms are not well conserved, upper arrow). **c** The initial injury response between species is very similar, but variability increases for mechanisms associated with cell activation, cell cycle progression, blastema formation, and morphogenesis, and then becomes more similar again during the differentiation and scaling phases of regeneration. The hypothesis represented in **d** asserts that variation between species is relatively large for processes that occur during wound healing, but low in the processes involved in blastema formation; after blastema formation, process variability may remain low (conserved) or increase (divergent).

prescribed times based on a specific developmental plan. Although it is also difficult to ascribe an initiating event in organ development, developmental biologists largely agree that the formation of a tissue anlagen or primordium is associated with precursor cells that are competent to receive inductive signals. In response to inductive signals (cell-autonomous or non-cell-autonomous), precursor cells launch a developmental program and self-organize into tissues comprising a diverse array of differentiated cell types. Thus, it may be appropriate to consider the beginning of regeneration per se as the point when injury-activated cells accumulate at the injury site and adopt a development-like state (Table 1). This would align the beginning of regeneration with anlagen formation and distinguish wound healing from events specific to regeneration (Fig. 2c).

In support of this proposition, data from salamanders suggests that injury-induced cell accumulation is necessary but insufficient for regeneration (reviewed in ref. [23]). Across countless experiments, researchers performed surgical interventions to study how nerve-secreted signals regulate the regenerative response in ambystomid salamanders and newt limbs[24,25]. These results show that wound healing and proliferative cell accumulation occur after denervating a limb, but aggregated cells fail to progress to morphogenesis. In further support of this idea, inhibiting certain signaling pathways (e.g., Wnt and Hedgehog signaling) allows

damaged tissue to repair, but regeneration does not occur[26–28]. Together, these findings support the concept that the ability to accumulate proliferative cells is a necessary but insufficient feature that distinguishes wound healing processes from regeneration. Instead, it suggests the transition to regeneration is characterized by proliferative cells acquiring the ability to undergo patterning, differentiation, and growth, a state often referred to as blastema formation[29]. However, data linking signaling pathways and nerves to regenerative competence also points to the existence of other essential cellular and molecular events that are required for regeneration to begin in a robust and recognizable way[26–28]. Thus, while the blastema is a common element of what may represent an evolutionarily conserved regenerative feature, it also highlights the critical need to determine which specific cellular and molecular characteristics of the local cellular environment are essential for regeneration to proceed.

The discussion of molecular signals that initiate regeneration evokes broader questions about what initiates the start of regeneration. For example, how is the amount of loss or damage requiring regeneration detected? Are specific regenerative programs only initiated after significant cell loss? If so, what are the mechanisms that activate them? There are many unknowns regarding injury-sensing mechanisms. Are there particular

---

**Box 2 ▮ Regeneration, tissue renewal, and all things in between**

When scientists use the term *regeneration*, they do not always distinguish between processes that share similar features or outcomes, making it difficult for researchers outside the field to understand the significance of a given experiment. For example, it has become popular for researchers to conduct "regeneration" experiments on amphibian and fish embryos or at different stages of imaginal disc development as a proxy for studying regeneration in adult animals[99–101]. However, most regeneration biologists agree that tissue repair in embryos reflects tissue restoration via *embryonic regulation*[102,103]. While experimental results using these models may provide insight into how specific signaling pathways respond to cell damage or loss, such embryonic "regeneration" is often restricted to an early developmental window when tissue morphogenesis is still ongoing and should not reflect the regenerative capacity of that species' fully differentiated tissue. Moreover, we cannot assume the regulatory mechanisms used to rebuild tissue in embryos are the same as those needed to restore developmental potency to adult cells (see Box 3).

Those studying regeneration in animals and plants generally use the term "regeneration" to refer to *reparative* regeneration: the faithful replacement of mature tissues, organs, or body parts in response to injury to restore the original structure and function. However, Thomas Hunt Morgan specified two modes of reparative regeneration that were not sharply separated: epimorphosis and morphallaxis. He defined epimorphosis as that mode where the "*proliferation of material precedes the development of the new part*" and morphallaxis as the mode "*...in which a part is transformed directly into a new organism or part of an organism without proliferation at the cut-surfaces*"[36]. Although some researchers have argued that a strict division does not exist between these two modes[104,105], most agree that epimorphosis is likely occurring in most regenerative organisms. In contrast, morphallaxis might be restricted to specific species or tissues (e.g., in the planarian intestine[106]). Animals such as flatworms deploy both modes in that a mass of new, proliferative tissue accumulates at the injury site prior to regeneration but cell re-arrangements also occur to integrate old and new cells and complete the regenerative process[107,108]. Regardless, these terms remain useful when discussing complex tissue or organ regeneration in response to injury. The applicability of "epimorphic" regeneration becomes less clear when our attention turns to examples such as muscle or hair follicle replacement in mammals (popular *in vivo* models also referred to as regenerative phenomena). In fact, these examples and others wherein dedicated multipotent stem cells underpin the turnover of single lineage tissues (e.g., feathers, gastrointestinal lining, blood, etc.) were historically referred to as "physiological regeneration" and are more aptly examples of tissue *homeostasis* or *renewal*. Far from being unique, this type of tissue "regeneration" is ubiquitous among almost all multicellular eukaryotes, in contrast to the epimorphic regeneration capacity of complex tissues, organs, and body parts in select animals. Thus, we should be cautious when equating ubiquitous, homeostatic phenomena to those with a more restricted phylogenetic distribution and a spontaneous, irregular starting point (i.e., injury-induced epimorphic regeneration).

But what happens when our neatly divided paradigms collide? For example, invertebrates such as planarians and *Hydra* exhibit high tissue turnover and almost unlimited regenerative capacity (i.e., whole-body regeneration). Although their response to injury features hallmarks of epimorphic regeneration, including the accumulation of proliferating cells at the wound site, they also maintain pluripotent somatic cells that constantly replenish the entire animal, such that all their tissues (including those that are not typically replenished in vertebrates, like the central nervous system) are in a perpetual state of renewal. Nevertheless, such animals provide an opportunity for studying the intersection of regeneration, homeostatic tissue renewal, and repair as they apply to all organisms. These examples underscore how a term like regeneration can be used in reference to functionally different processes, even though the differences may seem nuanced to those outside the field. This is especially evident when defining the basic component processes that comprise regeneration (Table 1) and determining the degree to which examples of regeneration in diverse species (or life stages) represent convergent or homologous events.

---

**Box 3 ▮ Where do developmental and regenerative processes most overlap?**

In reconsidering regeneration as a set of fundamental processes starting with wound healing, it becomes apparent that to say regeneration merely recapitulates development is an oversimplification. Since early animal regeneration experiments and evidentiary support of cell theory swept aside preformationist notions of development, a major point of inquiry has been the degree to which development and regeneration represent similar events deployed during different life stages. Historically, the relationship between development and regeneration put one in service of the other, depending on the scientific era. For example, 19th century embryologists studied regeneration to better understand developmental processes. However, late 20th century technological advances for assessing gene expression and function provided the means for studying genetic interactions during embryonic development directly in the embryo. The rise of genetic model organisms to study development then, in turn, created both an opportunity to leverage these research models for studying regeneration and a framework to uncover the genetic basis for regeneration, an approach that still dominates the field today.

This more recent mode of inquiry has operated under the assumption that the evolution of the embryo predates the evolution of regeneration, and thus, researchers contextualize their studies by asking what developmental pathways are redeployed during regeneration. On the contrary, might regeneration have provided the molecular building blocks and genetic circuits for embryonic development? Observations made by a number of experimental embryologists have hinted that patterning processes regulating regeneration in metazoan embryos were already present in early unicellular organisms[36,109,110]. If viewed through contemporary molecular biology and genetics, it is possible that genetic circuits necessary to restore patterns in the first multicellular organisms were later co-opted to help build the embryo. While the evolution of molecular mechanisms underlying regeneration and embryonic development may echo the conundrum of the chicken or the egg, considering alternative hypotheses about their relationship has value for understanding how regenerative ability is missing or curtailed in some animals.

---

molecules or physiological changes communicating the amount or extent of tissue damage? Does the loss of cell-cell contacts act additively or in parallel to chemical sensing mechanisms? Does the new interaction of cells from different anatomical positions stimulate the regenerative program[30–32]? Overall, the common theme that emerges from these questions is the need to expand data collection on regenerative ability within and across species, particularly by adding data from more examples of failed or intermediate regenerative success. More comprehensive data will allow researchers to better identify cellular, molecular, and functional commonalities in successful regenerative processes versus unsuccessful ones.

### What constitutes the end of a regenerative process?

Indisputably, the most desirable outcome of a regeneration process is to regain tissue shape and function. Thus, understanding and defining when and how a regenerative process restores a functional state is equally important to determining how it starts.

---

**Box 4 | The concept of de-differentiation**

Despite being one of the most common terms used by the regeneration community, de-differentiation remains ambiguously defined conceptually and experimentally. When used in the context of a cell participating in regeneration, de-differentiation originally referred to a loss of the terminally differentiated state in favor of a reacquired capacity to proliferate (Table 1). Elizabeth Hay cautioned about the extension of de-differentiation to include expanded lineage plasticity anticipating that it would lead to terminological chaos[111]. However, many contemporary biologists extend the definition to imply the loss of a stable phenotype (or identity) and the reacquisition of cellular plasticity[112] (i.e., differentiation potential). Experimental evidence for the extended definition of de-differentiation, as defined here, is supported in part by studies leveraging scRNA-seq comparisons of tissue undergoing development versus those undergoing regeneration, where cells undergoing regeneration begin to resemble embryonic cells found in the developmental anlagen[14]. Conceptually, a useful metaphor revises Waddington's epigenetic landscape model (where cells were originally envisioned to move downhill only) to accommodate cellular re-programming, as demonstrated in larval *Xenopus* cells and later by using the Yamanaka factors (Oct4, Sox2, Klf2, and c-Myc) to re-program somatic cells in mammals where cells are now known to possess the potential to move "back up the hill" and de-differentiate to a previously occupied state[113-116]. Is the modern definition of de-differentiation akin to cellular re-programming (an important, if semantic, distinction)? If de-differentiation is instead partial re-programming, how do cells partially re-program naturally? And are there specific regulatory factors that limit complete reversion to a pluripotent state? Does de-differentiation always imply at least partial cellular re-programming, or trans-differentiation, in which cells gain the potential to switch fates? These questions should challenge us to consider defining de-differentiation more precisely, including how it does or does not differ from re-programming. In doing so, we can ask functionally relevant questions. Which cell types undergo de-differentiation? Do multiple cell types have the capacity to de-differentiate in different contexts, or are there specialized cells that have the capacity to respond to injury signals and act as progenitor cells?

Another perplexing question is whether cells in animals with pluripotent somatic cells can (or ever) undergo de-differentiation. As evolution gave way to the germline-somatic division, did multiple differentiation strategies arise? For example, did one pathway lead to the evolution of adult stem cells (e.g., satellite cells, intestinal crypt cells, hematopoietic stem cells) while another evolved multipotent progenitors (e.g., pericytes and fibroblasts) as a means for tissue renewal? These and other questions could be tested experimentally by combining lineage tracing, single-cell dissociation, next-generation sequencing, and genomic profiling before and after injury and over sufficient time scales.

---

Most descriptions of tissue or whole-body regeneration indicate that repair processes shift to a remodeling and growth/scaling phase after the initial expansion of cellular material and patterning[33]. But exactly how regeneration restores form and function is not clearly defined. For example, as cells accumulate and proliferate, how do they appropriately regulate pathways that terminate cell cycle progression and how is this balanced with tissue scaling? Work on the Hippo pathway has demonstrated that its signals can be modulated to regulate growth during development[34]. A similar use of this pathway likely controls growth during regeneration[34,35]. However, precisely clarifying when regenerative healing and tissue restoration shift to a growth phase and identifying the molecular mechanisms that regulate this transition is a major objective for our field.

One possibility is that the end of regeneration recapitulates the patterning and growth of embryonic organs, where growth ultimately becomes regulated at the organismal level. For example, regeneration in molting animals (e.g., crustaceans) produces a miniature facsimile of the original appendage, which is only capable of additional growth upon subsequent molts[36-38]. In zebrafish, caudal fin regeneration is generally consistent with the overall size of the animal, but genetic mutants do exist in which regenerating fins lose their allometry and produce dramatically overgrown fins[39]. Nonetheless, regeneration generally restores missing tissues, their structures, and their functions, suggesting that although the precise mechanisms underlying the integration of new and existing tissue may vary across species, they achieve a common output.

In trying to understand why regeneration is absent in some animal species or tissue types, one hypothesis is that regeneration initiators are missing or not sufficiently activated, while another suggests that some animals lack the capacity to create restorative cell states. Another hypothesis is that mechanisms that normally terminate phases of regenerative healing have been co-opted to inhibit regeneration completely. For example, the evolution of "molecular brakes" in some animals may have rendered certain tissues regeneration-incompetent (e.g., heart muscle and the auditory epithelium)[40]. The acquisition of molecular breaks is also observed within an animal's lifetime: neonatal mammals can regenerate a subset of tissues, including cardiac muscle[41]. Notably, cardiac muscle has been observed to regenerate in newborn mice

for approximately one week before this ability is lost[42]. Could this apparent loss of regenerative ability as the animal ages result from inhibitory mechanisms that halt a specific component process? Comparisons across regenerative versus non-regenerative species/tissues are likely to assist in exploring this hypothesis[29]. In addition, a greater emphasis on pinpointing mechanisms that terminate or inhibit regenerative processes in cases where regeneration appears to be interrupted or halted after successful initiation may have profound implications for understanding how reduced regenerative capacity evolved in certain animal lineages.

Notably, unbridled overgrowth of regenerating tissue is rarely observed, supporting the hypothesis that regenerative species have mechanisms that provide tight control over proliferation and morphogenesis. Furthermore, reports of tumors in highly regenerative species are uncommon in the literature, although not unknown (reviewed in[43,44]). Neoplasms can be induced in newts[45] and zebrafish (reviewed in[46]), and epidermal neoplasms occur with some frequency in captive salamander colonies (see "De-Mystifying Salamander Cancers" Facebook group). In spite of these observations, the idea that highly regenerative organisms live tumor-free and are resistant to developing cancer remains a widely held belief[43,47]. Interestingly, epimorphic regeneration often requires highly conserved pathways that are associated with cancer, and disrupting tumor suppressor genes such as Hippo, p53, or PTEN in planarians and zebrafish can occasionally lead to the formation of tumor-like structures[48-51]. Thus, while the incidence of significant, bona fide tumorigenesis in highly regenerative animals appears low, it seems likely that pathways commonly dysregulated in human cancer are tightly controlled during regenerative healing[52-55]. Alternatively, the expansion of tumor suppressor gene families in mammals may correlate with their reduced ability to regenerate[54]. For example, the human ARF protein produced as an alternately spliced gene product from the p16 locus appears to be absent from the genomes of many regenerative species[54]. In addition, injecting newt myotubes with a plasmid encoding the human tumor suppressor p16$^{INK4a}$ blocks de-differentiation and cell cycle re-entry[56]. Therefore, tumor suppressor genes may represent another layer of complexity that negatively regulates regenerative responses in mammals. Unraveling the functional relationship between tumor

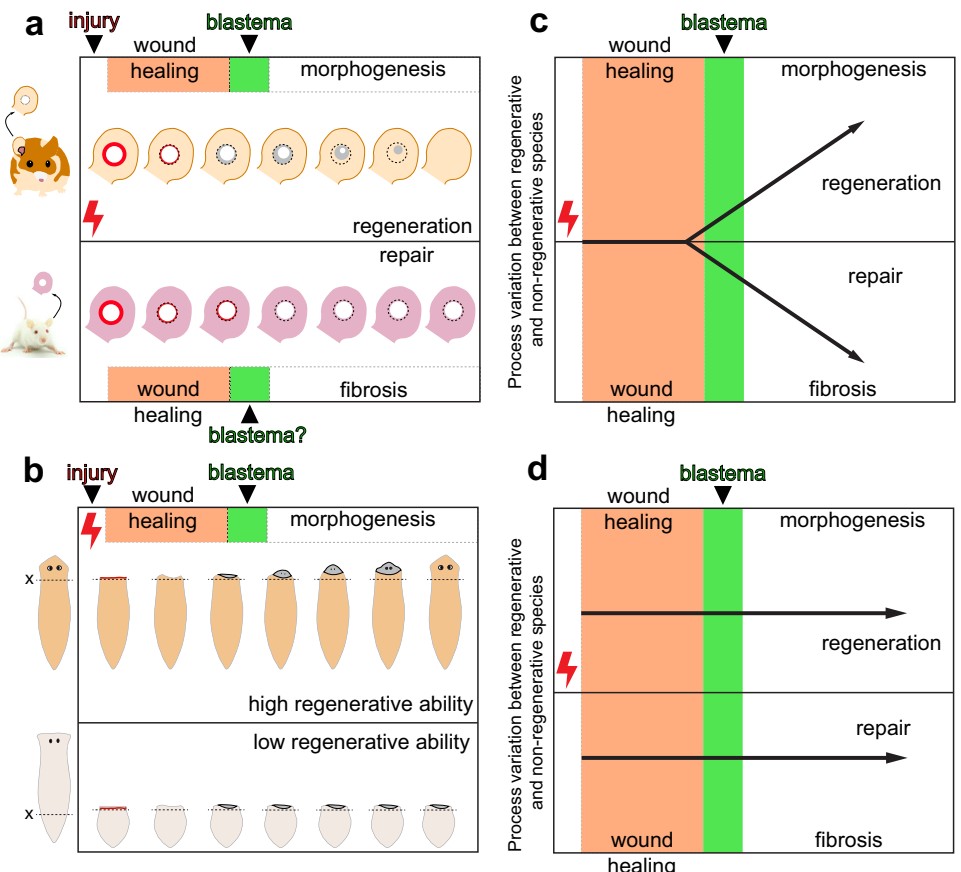

**Fig. 2 Alternative hypotheses describing scenarios for observed differences in regenerative ability. a, b** Examples of variation in regenerative ability across species. **a** Tissue healing is quite different between closely related (~18 mya) *Acomys cahirinus* (spiny mouse) and *Mus musculus* (laboratory mouse; Adobe Stock Image used with educational license) where *Acomys* exhibit complex tissue regeneration in the ear pinna and identical injuries heal with scar tissue (and no regeneration) in *Mus*. **b** While two flatworm species *Schmidtea mediterranea* (orange) and *Dendrocoelum lacteum* (gray) are capable of head regeneration, *D. lacteum* exhibits poor head regeneration from posterior fragments (*S. mediterranea* has near limitless regenerative ability from any fragment). **c** Comparing regeneration and fibrotic repair, the early events that occur prior to new tissue formation could be similar between species, only to diverge as mechanisms specific to regeneration or fibrotic repair are activated. In the example presented, this divergence occurs immediately prior to activation of a developmental state (i.e., blastema formation), although under this hypothesis it could occur later. **d** An alternative is that regenerative and fibrotic healing are evolutionarily distinct and thus upon injury two different healing trajectories, and their mechanistic underpinnings, are expressed.

suppression, cell cycle control, and regeneration will require either in vivo genome editing or the introduction of genetically edited cells. However, these tools are currently limited to only a few regenerative organisms.

### How does patterning happen across different scales to generate proportioned organs and tissues?

Understanding the mechanisms by which organisms set and restore size and shape is a formidable challenge in development and regeneration. Evidence from divergent organisms indicates that patterning and growth during regeneration depend on the activation of major components of developmental GRNs and signaling pathways. For example, a conserved role for Wnt signaling in re-establishing polarity during regeneration is supported by work in many regeneration systems, including *Hydra*, acoels, planarians, and amphibians[13,27,57,58]. Despite redeploying conserved developmental pathways, these processes need to operate on very different timescales and, in some cases, across orders of magnitude in scale, which remains difficult to reconcile with current knowledge about patterning in fields of cells.

In one illustration of how organisms can cope with scaling problems, *Drosophila* embryos balance cell proliferation and apoptosis to regulate proportion during development. Specifically, Bicoid protein is distributed in a gradient across the developing embryos and organizes anterior development[59,60]. Yet classic experiments revealed that increases in Bicoid dosage levels, which expand the anterior fate map of the embryo and should give rise to patterning defects, can produce normal larvae[60]. This restoration of proper proportion is driven by increased cell death in the expanded anterior regions, suggesting that the embryos have mechanisms for sensing and adjusting the relative proportion of anterior fated cells[61]. Cell death has also been implicated as a key process required in *Hydra*, planarian, and mammalian liver regeneration[62]. But how do cells detect when there are too few or too many cells and adjust the rate of proliferation or cell death? How do they maintain the correct ratio of specific cell types, particularly in a replacement tissue that must match the scale of the existing animal? Clearly, these are complex, challenging problems that are relevant to both development and regeneration, making experiments in one context informative for the other.

Besides balancing cell proliferation and death to achieve a proper number of building blocks, injured tissues must specify the fates of new cells and rearrange them appropriately to establish tissue size, proportion, and function. One historical view of how this is accomplished is that cells carry information in a Cartesian code to

interpret positional details, and this code is deployed in response to morphogen gradients for pattern formation[63]. A molecular version of this theory is that cells have different chromatin and transcriptomic states based on their developmental history and can both emit and respond to signals in their local environment. Therefore, overall patterning can vary depending on the type or strength of the signals present and the state of cells interpreting the morphogenic cues. A striking example of the importance of this type of injury-induced communication is the dysregulation of canonical Wnt signaling in planarians, which disrupts anteroposterior polarity and leads to the regeneration of ectopic heads or tails[57]. In addition, the Wnt-signaling pathway also plays a role in regulating the proportion of new tissues, as disrupting the striatin-interacting phosphatase and kinase (STRIPAK) complex increases worm length by expanding the posterior *wnt1* signaling center and dysregulating axial scaling[64].

Important regeneration and growth signals are likely not restricted to secreted ligands and well-known signaling pathways. They likely involve ECM components, biomechanical inputs, redox state fluctuations, and changes in metabolic states[65–68]. Ion sensing has also been linked to organ size and regeneration[69]. For example, the zebrafish mutant *longfin* exhibits fin overgrowth due to ectopic expression of the ion channel *Kcnh2a*[39]. Thus, ion sensing and regulation may be a genetically encoded and tunable mechanism for "reading" positional information and producing the correct amount of growth. However, we do not yet know what controls the strength of different types of signals in different contexts and how the regulatory pathways in development might differ from those activated during regeneration. Additionally, although particular signaling pathways may be conserved across contexts, the exact cellular and molecular mechanisms in which they are deployed may differ depending on the size, proportion, and types of tissue being replaced. To this end, embryos typically develop on a much smaller scale than the replacement tissue that is produced during restorative regeneration, so while embryonic pathways may be re-used during regeneration, there are likely key mechanistic differences in how they are employed during the latter.

## Is regeneration driven by gene regulatory modules that are conserved across species?

One rapidly expanding area of interest in regeneration biology is the application of genomic tools to understand how gene expression is regulated over time, in different injury and tissue contexts, and between species. Many recent studies have focused on identifying regulatory elements, particularly enhancers, that are potentially unique or specifically activated after injury in regenerative organisms/tissues (see Box 5). These efforts are rational extensions of studies showing that enhancers are sites of dynamic genome interactions with gene promoters during cell fate transitions in various developmental contexts and organisms[70–74]. Further, the genetic tools and cell culture systems used routinely in *Drosophila* and ex vivo mammalian cells to dissect regulatory mechanisms are not yet optimized in most highly regenerative organisms. Thus, it is reasonable to use paradigms emerging from traditional models as the basis for targeted experiments in regenerative organisms as a starting point for uncovering gene regulatory modules that function during regeneration. However, given that the genome regulatory mechanisms that drive regenerative processes and transitions may be different from those operating in non-regenerative species, it is also critical for our field to expand our hypotheses and approaches beyond those that emerge from studies in common animal models. To ensure that we identify the key regulatory mechanisms, including potentially novel ones, that control essential

regenerative processes, we must invest the resources needed to customize problem-specific tools in regenerative models. For example, there are chemical biology tools (e.g., degrons, azide-labeled non-canonical amino acids) that are not commonly used in our fields but would allow us to address specific, mechanistic questions arising from decades of observational and functional studies.

Any individual group studying genome regulation during regeneration will likely find that evaluating all possible mechanisms of genome regulation is a formidable challenge. Thus, innovation and impactful discovery in this area will particularly benefit from coordinated interaction and collaboration across research groups so that experimental design and genomic data collection can be readily comparable. In addition, a broad view of genomic regulation across multiple regenerative species will allow us to identify conserved mechanisms, even if the specific genetic modules they activate or silence are different[13] (Fig. 3). Whether there is greater conservation of upstream genome regulatory mechanisms, the specific proteins orchestrating them, or the genetic modules they target is not likely to have a single or simple answer (Fig. 3). However, we can better distinguish which molecular mechanisms are functionally significant during regeneration by focusing on how specific genetic modules that are known to be activated and essential for regeneration are regulated in regenerative versus non-regenerative contexts.

For example, in the regeneration-competent zebrafish retina, expression of transcription factor *ascl1* is induced upon retinal injury and required for retinal regeneration[75,76]. Interestingly, Ascl1 is conserved in mice and required for their retinal development[77,78], but *Ascl1* expression is not induced after injury of the non-regenerative mouse retina[79,80]. This observation inspired experiments in which *Asl1* was ectopically expressed in explanted adult mouse retinas[81]. However, although Ascl1 functions like a pioneer factor in some ways (e.g., inducing transcription of some relevant target genes), ectopic *Ascl1* expression alone was insufficient to induce functional de-differentiation, proliferation, or redifferentiation in injured mouse retinas[81]. Yet, importantly, further experiments found that treatment of mouse retinas with ectopic *Ascl1* expression in combination with a histone deacetylase (HDAC) inhibitor (which broadly increases genome accessibility) is significantly more successful in activating productive de-differentiation, proliferation, and redifferentiation of mouse cells[82]. These results are exciting, as they suggest it is indeed possible to reactivate existing genes and induce regeneration in non-regenerative tissues[83]. Nevertheless, many unanswered questions remain: are HDACs facilitating Ascl1 binding to the genome, and if so, at which loci? Which of these newly bound Ascl1 loci are functionally important, and for which steps in the regenerative processes? Why is the induced regeneration of mouse retinas still less robust than the endogenous regeneration process in zebrafish? What regulatory mechanisms or targeted gene expression programs are missing (or inhibitory)? Experiments addressing these specific questions have the potential to both further our understanding of retinal regeneration and uncover widely conserved molecular mechanisms that regulate essential genetic modules in other regenerative contexts.

From a broader perspective, there are many interesting and unanswered questions regarding the role of genome structure and function in regenerative processes. For example, is it relevant that many regenerative organisms, including vertebrates (e.g., axolotls) and invertebrates (e.g., planarians), have large genomes containing significant amounts of repetitive sequence? What do the repetitive sequences signify? Are these sequences serving as regulatory platforms for transcription factor binding, or do they play more instructive roles in regulating cellular plasticity? Interestingly, studies in mammalian stem cells and early embryos

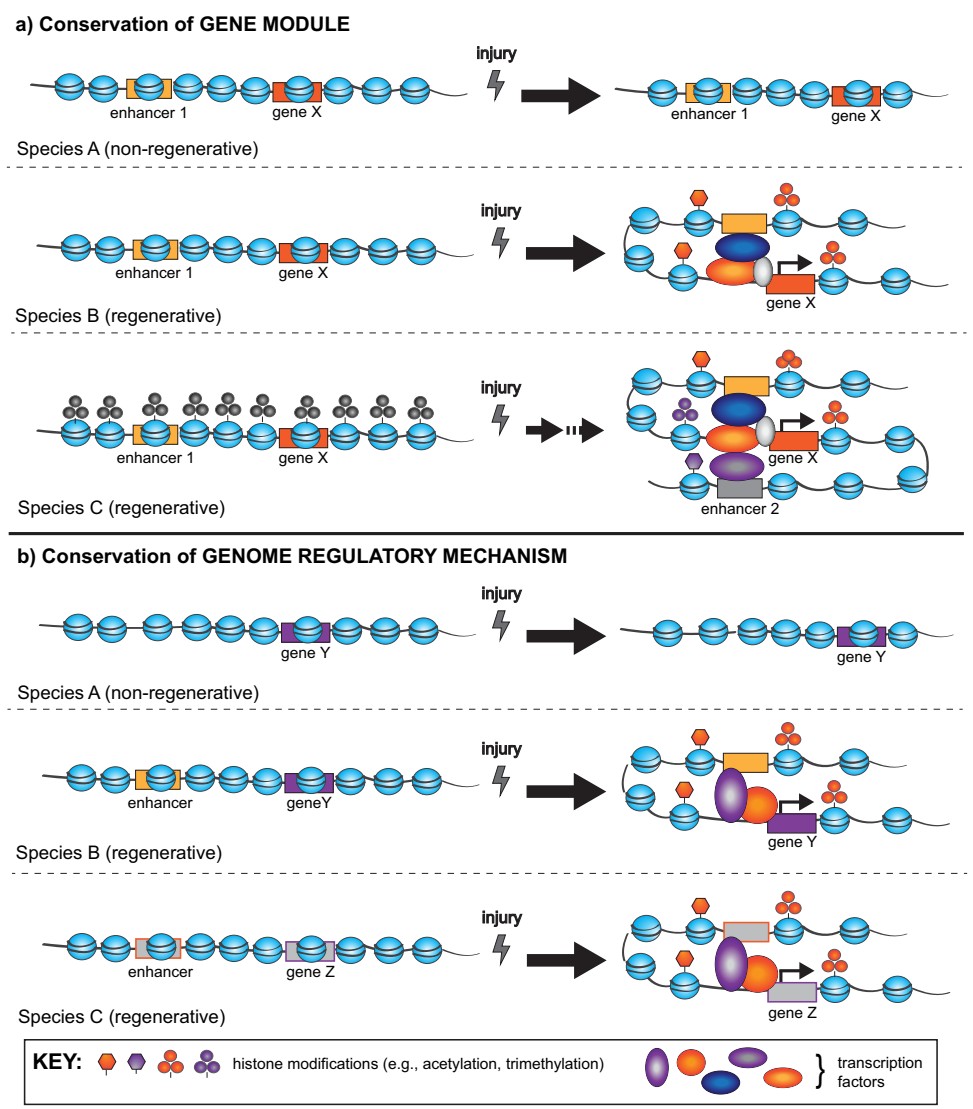

**Fig. 3 Hypothetical model illustrating conceptual ways genome regulation may facilitate regenerative capacity and regenerative processes. a** In this model, specific gene modules (e.g., coding genes and enhancers) are conserved among various species but regulated differently after injury. This model represents situations where specific genes (e.g., gene X) are present in multiple species, but only activated after injury in regenerative species. It also reflects the possibility that injury-induced genes are activated by different mechanisms in different regenerative species. **b** In this model, injury induces similar genome regulatory mechanisms (e.g., changes in histone modifications and the expression of pioneering transcription factors) but they have different functional outcomes due to evolutionary differences in the presence and arrangement of specific genetic modules (e.g., the enhancer is only present at this locus in regenerative organisms).

suggest that transient activation of repetitive regions plays a major role in regulating gene expression and chromatin state at critical developmental transitions[84–86]. Are similar mechanisms operating during cell state transitions after an injury? As with the other major knowledge gaps in our field, more comparative studies across species and developmental contexts will be needed to unravel the answers to these questions.

## Concluding remarks
One of the oldest and most enduring questions identified by regenerative and developmental biologists is why regenerative ability is unevenly distributed among metazoans[87]. Why, with over two centuries of regeneration research behind us and major technological advances in molecular biology, next-generation sequencing, and genetic engineering occurring at an ever-quickening pace, does this problem continue to challenge our field? First, evolutionary problems that span large phylogenetic distances are notoriously difficult to address experimentally, and debates about the adaptive nature of regenerative ability remain unresolved. Second, tackling this question necessitates expanding the diversity of species used in regeneration studies[88], which faces significant financial and practical barriers. Understandably, many scientists prefer to take advantage of the extensive toolkits available in a small subset of genetic model organisms and shy away from the risks associated with learning or developing a new model system. Of course, the decision to study a new model or species should be driven by scientific questions.

In our discussions at the Banbury workshop, there was broad consensus that comparative studies are essential to identifying the possible cell states, mechanisms, and functions critical for those processes outlined in Table 1. Unfortunately, experimental workflows designed for one species are rarely practical across multiple species due to various confounding factors (e.g., regeneration timing, anatomical differences, genome quality disparities, the need for species-specific expertise, etc.). Instead, the field should invest in collectively

---

**Box 5 | Challenges in identifying regeneration-specific enhancers**

Enhancers are non-coding regulatory elements in the genome that interact with gene loci to regulate their expression. A single enhancer typically contains recognition motifs for multiple transcription factors (TFs). It also requires the productive binding of multiple TFs for activation[117-119], a feature that facilitates the integration of various inputs (e.g., cell type-specific TFs and signaling-specific TFs) to create a new output. Studies in regenerative animals and tissues have uncovered enhancers that show increased chromatin accessibility after injury and whose increased openness correlates with the expression of wound-induced genes[120-126]. In some of these studies[122,123,125,126], activated enhancers were identified based on a gain of histone H3 lysine 27 acetylation (H3K27ac), a modification known to mark active enhancers across multiple organisms and contexts[127,128]. Recently, other studies have used the Assay for Transposase-Accessible Chromatin with high-throughput sequencing (ATAC-seq)[120,125] or tissue-specific transgenes expressing epitope-tagged histones[121] to detect wound-induced increases in genome accessibility or nucleosome turnover, respectively, at specific genomic loci. Together, these approaches are an important first step in addressing whether there are unique regulatory elements in the genomes of regenerative organisms and specific injury-activated mechanisms in these animals that can reactivate developmental GRNs (Fig. 3).

These pioneering studies raise important questions that require further investigation. First, are the methods used sufficient for identifying functional enhancers? Emerging data from the transcription field suggest that these approaches may only identify a small subset of functional enhancers due, in part, to the following considerations: (1) individual genes are often regulated by multiple enhancers[129-131], (2) enhancers identified by chromatin profiling do not correlate very well with assays of enhancer function[132] and, similarly, (3) not all injury-activated enhancers were found to be functionally required for regeneration[123]. Additional transgenic approaches will be needed to pinpoint injury-regulated enhancers that are functionally relevant and work in synergy with other genomic regulatory elements.

Second, how can we best identify the TFs that activate or are recruited to injury-regulated enhancers? Although sequence analysis of regulatory regions marked by specific chromatin modifications or increased accessibility is a valid and worthwhile first step, these algorithms depend on known TF recognition site information. As a result, they cannot identify the binding sites of novel TFs or even TFs with poorly characterized binding motifs. It is also very likely that regeneration enhancers will require the binding of multiple TFs to activate downstream genes specifically and robustly. In addition to using established omics methodologies, our field should optimize and develop biochemical approaches that address specific, outstanding questions about gene activation upon injury and during the regenerative response. Although such methods can be technically challenging, especially when starting with complex tissues versus cultured cells, the rewards would be high if they allowed the identification of relevant binding factors at essential gene loci. Third, what are the mechanisms that specifically activate these enhancers after an injury? For example, does it matter which histone acetyltransferase (i.e., p300 or CBP) acetylates a specific enhancer upon injury? Also, is H3K27ac even required for the activation and function of these enhancers? The latter question is particularly relevant given that enhancer H3K27ac is dispensable for activating gene expression in static cell culture but necessary for cell state transitions such as differentiation[133]. Another recent study reaffirmed that not all enhancers require the same core cofactors, such as the H3K27ac acetyltransferases p300 and CBP, to activate downstream target genes[134]. Although high conservation makes H3K27ac a convenient mark for identifying enhancer locus candidates, it will be necessary for our field to generate custom reagents that recognize the homologs of other cofactors, e.g., Mediator and BRD4, in regenerative species to identify those enhancers that are specifically regulated by these other complexes.

---

designing experimental pipelines that can be deployed across species in individual laboratories with high fidelity. Such pipelines would use existing technologies to generate data that may not be significant in any species but could synergize to generate and address many testable hypotheses. Multi-species studies can provide a platform to test the widely held position that a core regenerative program is conserved among metazoans, and its unequal distribution reflects loss and re-emergence in distant lineages. As the number of species used in regeneration research grows within taxonomic groups and across increasingly distant lineages, it will also provide an opportunity to rigorously examine if the alternative hypothesis may be true: that regeneration has independently evolved in numerous lineages. If the latter hypothesis was shown to be true, it would radically expand potential avenues to explore in regenerative therapies.

Adding to the complexity of discovering the mechanistic basis for interspecific differences in regenerative ability, cellular changes associated with aging (e.g., mutations, metabolism, epigenetic states) are gaining recognition as complementary problems with solutions that may help unlock the potential to stimulate regeneration in humans. Suggestively, animals with the capacity for whole-body regeneration, like *Hydra* and planarians, appear to be negligibly senescent[89], animals with indeterminate growth have high regenerative ability[90], and recent work in spiny mice suggests connective tissue cells from these regenerative mammals are highly resistant to stress-induced cellular senescence[91]. Moreover, long-lived animals and those with post-metamorphic life stages often lose or have diminished regenerative capacities in adulthood compared to their fetal, neonatal, or juvenile life stages[92]. While the cellular and molecular mechanisms underlying this loss remain poorly understood, possibilities include changes in how cells sense injury signals, cell-autonomous features that prevent cell activation or cell state alternations, or a decline in homeostatic cellular and tissue renewal (Table 1 and Box 2). Together, it is increasingly clear that major advances in dissecting the molecular logic of regeneration (and reconstructing it in humans) cannot occur by studying a handful of organisms. Instead, expanding the collective effort of many research labs across an increasing diversity of organisms can provide answers to some enduring questions in animal biology while potentially unlocking new breakthroughs in human regenerative medicine. Excellent examples of such efforts include comparative studies contrasting species of salamanders, fish, or flatworms[8,93-96], which revealed cellular and molecular insights.

Of course, science at the scale we suggest requires coordination across organisms, labs, and institutions, not to mention a financial strategy to support such activities. Additionally, there is a need for investigators to invest in multiple approaches to rigorously test conclusions on which subsequent studies are based. Rigor and reproducibility, once a cornerstone of the scientific method, have taken a distant backseat to novelty[97]. Potentially insightful work demonstrated in one model system, or species is strengthened, not weakened, through repeated experimental testing by multiple groups and through testing in other regenerative organisms[98]. We must ask this of our field. Cross-species study reproduction and hypothesis testing provide another dimension related to the conservation and canalization of regenerative mechanisms. To understand the fundamental principles of regeneration, we urge our peers and other scientists to revisit some of the most enduring questions in our field.

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

## Acknowledgements

We thank the Banbury Center for providing an amazing atmosphere for our vintage workshop experiment and, specifically, Rebecca Leshan for her backing and encouragement. We acknowledge Genentech and The Cold Spring Harbor Laboratory Corporate Sponsor Program for supporting the workshop. In addition to input from all attendees of the Banbury Workshop that informed this perspective article, we thank Francesca Mariani for significant editorial input. In a better-designed version of itself, academia would encourage the type of collaborative effort that went into this paper to be formally recognized with co-authorship. While nothing precludes this arrangement, how current conflict of interest (COI) declarations impact review procedures for grant proposals and promotion panels created unavoidable conflicts among participants to be co-authors. Lastly, we would like to thank John (Jack) Allen and Kelly Ross for their perspectives on an early version of this paper. Research in A.W. Seifert's lab is funded by NIH grants R01 AR070313, R21 DE028070, the ASAP Collaborative Research Network through the Michael J. Fox Foundation, and DOD CDMRP 6W81XWH2110503. Research in R.M. Zayas's lab is funded by NIH grant R01 GM135657. Research in E.M. Duncan's lab is funded through NIH grant R35 GM142679.

## Author contributions

A.W.S., E.M.D., and R.M.Z. recorded detailed notes of discussions from the Banbury Workshop (Box 1). A.W.S., E.M.D., and R.M.Z. used the compiled notes to draft the manuscript. R.M.Z. incorporated suggested edits from meeting participants into the drafted manuscript. A.W.S. and E.M.D. created the illustrations with input from R.M.Z. A.W.S., E.M.D., and R.M.Z. revised the document and discussed all major revisions.

## Competing interests

The authors declare no competing interests.
