## [Peer Review File · Communications Biology]

Reviewers' comments:

Reviewer #1 (Remarks to the Author):

This manuscript is neither a workshop report nor a review of the regeneration field. It is in essence an opinion piece largely of the views of the three authors but yet claims to be a report of discussions of all participants at the workshop, if this is the case and this is what it is supposed to represent then all participants should be listed as authors.

Overall the piece is well-written but very subjective and really represents the opinions and interpretations of the field of regeneration biology of authors.

If the authors want to represent more what is seen in the literature then several things in the piece need to be adjusted, for example:

1. lines 71-73, most people in the regeneration biology community consider wound healing to be part of regeneration and it is worked on in many model organisms. In fact wound healing and regeneration people have had many conversations and have actively decided that these are not separate processes or fields of study, wound healing is part of regeneration. This opinion is reiterated at several points during the manuscript (lines 168-172) and many will feel this an inaccurate interpretation of the field.

2. line 101-103, this is a subjective narrow definition of the term de-differentiation and does not encompass what the majority of people in regeneration and stem cell fields refer to as dedifferentiation.

3. Table 1 should include references and examples of what this group considers testable hypotheses for each of these biological processes.

4. lines 223-225, more recent work on crustaceans and regeneration should be considered and cited here, for example the work from the Averof lab on Parhyale limb regeneration.

5. The section of cancer and regeneration, lines 247,266 lacks many of the relevant references work from Tsonis P and Brockes JP, among others should be referenced in this section.

Reviewer #2 (Remarks to the Author):

The manuscript represents a very timely synthesis of the open questions in regenerative biology and provides a framework for how these questions might be addressed. I believe this work will be highly valuable for the field.

The manuscript is a result of a workshop attended by many of the leading figures in the regenerative biology, and formally reviewed by all participants. As such, this is an excellent work.

I have only one comment. In Box 1 (lines 708 – 727) physiological regeneration is referred to as “examples of tissue rejuvenation”. I was surprised to read this, as, at least to me, rejuvenation specifically means restoration of a youthful state, and this is how it is usually used in the context of stem cell research and aging. The authors equal homeostatic phenomena of physiological regeneration with rejuvenation, and I think this is not necessarily always the case. And indeed, whether and when regeneration leads to rejuvenation (in the sense of returning to a youthful, more functional state) is still an open question. If I’m wrong, it would be good to provide references in this section supporting the authors statement equaling physiological regeneration with rejuvenation. But my suggestion here is simply to distinguish physiological and reparative regeneration and not invoke rejuvenation.

Furthermore, multiple references missing some information (e.g. page numbers), and Kunh 1969 (line 34) is not in the reference list. But I believe these technicalities will be fixed during production.

Reviewer #3 (Remarks to the Author):

In this manuscript Seifert, Duncan and Zayas share insights from a recent workshop that took place at the Banbury Center of Cold Spring Harbor, where big and open questions in the field of regenerative biology have been discussed. In an effort to raise these questions logically, the authors categorically described the key steps common to regenerative process and posed different scenarios that could explain differences between regenerative and non-regenerative species when it comes to injury response and the outcome. Additionally, the authors highlighted the importance of increasing the diversity of model organisms used in regenerative research and enhancing cooperation between experts from different fields in an attempt to find answers.

First of all, I would like to thank the authors for putting this insightful piece together and allowing the missing members of the community to join in on the discussion. There is no doubt that such a discussion is very timely in light of all the new techniques that became available to the regenerative biology field recently.

Detailed comments:

Strengths

- 1-Excellent flow that keeps the reader engaged throughout.
- 2-The choice to focus on common questions related to regenerative biology rather than putting heavy emphases on a specific model organism or tissue.
- 3-High quality and depth of "known unknowns" discussed.
- 4-The content of Box 1-4 (very informative) and the experimental ideas in these sections.

Suggestions for improvement

5- While the quality and the context of the discussed questions are excellent, the obstacles limiting our progress in answering these questions are not crystallized. I would recommend the authors to clarify these obstacles more clearly. Are these obstacles merely due to technical limitations? Can we overcome these limitations using the modern technologies that the authors alluded to? As it is written it comes across as if it would be sufficient just to study a more diverse cohort of species. However, we are still lacking for example temporal resolution to capture mechanistic events happening in short time scale due to lack of imaging modalities that can longitudinally capture these events live in deep tissue over the course of many days that is required for certain examples of tissue regeneration.

6-In the abstract and introduction, the authors emphasize the importance of bringing together scientists and experts from various fields and describe the current workshop as a platform where this was achieved. However, the outcome is unclear to the reader. Could the authors please elaborate on key realizations that were made as a result of having technical experts together with developmental/evolutionary biologists?

7-One of the main take away messages is to increase the diversity of species used in research to enable comparisons that might answer some of the questions posed in the meeting. There are already excellent examples in the literature where researchers compared the regenerative process between different species (for example: PMID: 28632131, PMID: 24268695, PMID: 30462998). Could the authors elaborate on whether these comparisons have led to the expected outcome? And if not, why? Also, I assume that finding answers to the questions posed here requires coordinated action of many labs, perhaps in the form of a consortium, to ensure experiments are performed in a way where results are openly available and comparable (consistency in experimental design, technological

platform used, bioinformatic analysis parameters etc). The authors partially touch upon this in Line 348-351. In this sense, this article would be an excellent starting point for discussion. Was there any suggestion in the meeting on how to handle this?

8-Figures are not very self-explanatory. Please see more detailed comments below. Additionally, it might be helpful to have an extra column in Table 1 to clarify which steps belong to a) healing, b) blastema and c) morphogenesis.

Minor comments:

9- It might be wise for the authors to include a description of blastema in a Box. Do we consider formation of a blastema in internal organ regeneration such as the regeneration of the heart or spinal cord etc.?

10- The authors write that the conference attendees worked to formulate an ambitious plan to discover answers to new and enduring questions (Line 49-50). It is not clear what the plan is.

11- Line 76: It would be nice to spell out what the new research organisms are for the readers who might not be as familiar.

12- Line 176: Newts are salamanders. Please revise to salamander limbs or detail further as salamanders, such as axolotl and newts or use urodele amphibians as in other parts of the manuscript.

13- Line 181: The statement that "Collectively, these findings support the concept that the ability to accumulate cycling cells is a key feature that distinguishes wound healing processes from regeneration." does not fit in where it currently is as the authors have just cited studies that showed that accumulation of cycling cells is not sufficient in the absence of certain signals.

14- Line 412-414: The authors write "As researchers, individually and collectively, we believe progress can be made toward understanding regenerative ability in the context of evolution by designing experiments to solve some of the problems presented above." Could they expand on this through discussion of how new technologies and cross-disciplinary approaches might help in the design of these experiments? Were there any concrete suggestions within the group?

15- Line 423-424: could the authors please add references?

16- Line 423-427: maybe should not be in concluding remarks but in the introduction?

17- Line 817: Please write gene regulatory networks (GRNs).

18- Figure 1: Panel B-E, please revise the Y axis label to "Process variation between regenerative species".

19- Figure 2: Panel C and D, please revise the Y axis label to "Process similarity between regenerative and non-regenerative species or species" or similar.

20- Figure 3: Panel A, in the figure legends the authors write "specific genes are present and possibly activated during development across multiple species, but only induced after injury in regenerative species." The cartoon does not reflect this statement. I suggest that the authors to add a sentence such as "potential developmental gene" to the figure. Additionally, in the figure the authors display

differential epigenetic modifications that might be involved in gene regulation. It might be good to explain this in the figure legend.

21- Figure 3: panel B, is gene Z supposed to be a regeneration specific gene?

Response to Reviewers for MS: COMMSBIO-23-2022

We appreciate the invitation for us to revise our manuscript and concur with the decision to publish it as a Perspective. We address the reviewer comments point-by-point below and have highlighted our responses in **RED**. We have also highlighted relevant text in **RED** in our revised manuscript where we made changes to address points raised by the reviewers.

Reviewer #1

This manuscript is neither a workshop report nor a review of the regeneration field. It is in essence an opinion piece largely of the views of the three authors but yet claims to be a report of discussions of all participants at the workshop, if this is the case and this is what it is supposed to represent then all participants should be listed as authors. Overall the piece is well-written but very subjective and really represents the opinions and interpretations of the field of regeneration biology of authors.

We have addressed the major concern expressed here about the nature of the article by following the Editor's suggestion that we revise the manuscript as a Perspective piece rather than a review. We also added a box (Box1) outlining who attended the workshop, how the meeting contributed to the ideas contained within this Perspective, and indicating that the views expressed are endorsed by all attendees. We initially intended for all attendees to be authors of this piece but, unfortunately, several attendees raised concerns about how co-authorship would negatively impact grant and paper reviews in light of current COI guidelines at the NSF and many journals. Additional concern was raised about conflicts for senior scientists writing tenure letters for junior co-authors. For all of these reasons, the group asked the three of us who organized the workshop to draft and publish the article as sole authors.

If the authors want to represent more what is seen in the literature then several things in the piece need to be adjusted, for example:

1. lines 71-73, most people in the regeneration biology community consider wound healing to be part of regeneration and it is worked on in many model organisms. In fact wound healing and regeneration people have had many conversations and have actively decided that these are not separate processes or fields of study, wound healing is part of regeneration. This opinion is reiterated at several points during the manuscript (lines 168-172) and many will feel this an inaccurate interpretation of the field.

We are unclear about the point the reviewer is making here, as we believe we have presented wound healing/repair as a phase of the overall regenerative process throughout the manuscript (i.e., in Table 1, sections II and III, Figures 1 and 2). We respectfully disagree with the Reviewer that the contents of the article are inaccurate. We also note that this piece was read by twenty colleagues in the field, whose constructive criticism was incorporated into the review, so the views here represent a broader opinion than the three authors. Moreover, while it is broadly agreed that the early phases of regeneration and wound healing share many features at the process level, the outcomes are obviously different. However, to make sure this view is clear to our readers, we have adjusted the text (lines 57-62) and made modifications to both **Table 1** and **Figure 1** emphasizing our view that wound healing is an inseparable part of regeneration. We think this view is in line with that of Reviewer #1 and will satisfy their critique without stifling discussion and inquiry into how these processes differ between regenerative and non-regenerative species.

2. line 101-103, this is a subjective narrow definition of the term de-differentiation and does not encompass what the majority of people in regeneration and stem cell fields refer to as dedifferentiation.

We understand that this footnote to Table 1 could be confusing and have edited it accordingly (lines 90-92). The definition in the footnote is aligned with the historical/textbook definition of cell de-differentiation. Moreover, Box 4 (lines 869-903) presents the historical and contemporary definitions of de-differentiation, including an extension of the definition that equates de-differentiation to increased cellular plasticity.

3. Table 1 should include references and examples of what this group considers testable hypotheses for each of these biological processes.

We appreciate this suggestion. We endeavored to make the article accessible to a broad audience and keep the table straightforward. However, we provided many examples of testable hypotheses throughout the text and have further clarified that in this revision. For example, see lines: 108-113, 124-131, 230-234, 420-426, and 896-903.

4. lines 223-225, more recent work on crustaceans and regeneration should be considered and cited here, for example the work from the Averof lab on Parhyale limb regeneration.

Done (now lines 221-223, ref 37 and 38). Thank you!

5. The section of cancer and regeneration, lines 247,266 lacks many of the relevant references work from Tsonis P and Brockes JP, among others should be referenced in this section.

We have added these references (refs 45 and 56 within refs 43-56 pertaining to this topic and referenced in lines 246-264). Thank you.

Reviewer #2

The manuscript represents a very timely synthesis of the open questions in regenerative biology and provides a framework for how these questions might be addressed. I believe this work will be highly valuable for the field.

The manuscript is a result of a workshop attended by many of the leading figures in the regenerative biology, and formally reviewed by all participants. As such, this is an excellent work.

I have only one comment. In Box 1 (lines 708 – 727) physiological regeneration is referred to as “examples of tissue rejuvenation”. I was surprised to read this, as, at least to me, rejuvenation specifically means restoration of a youthful state, and this is how it is usually used in the context of stem cell research and aging. The authors equal homeostatic phenomena of physiological regeneration with rejuvenation, and I think this is not necessarily always the case. And indeed, whether and when regeneration leads to rejuvenation (in the sense of returning to a youthful, more functional state) is still an open question. If I’m wrong, it would be good to provide references in this section supporting the authors statement equating physiological regeneration with rejuvenation. But my suggestion here is simply to distinguish physiological and reparative regeneration and not invoke rejuvenation.

We thank the reviewer for pointing out that the terminology may cause confusion. We have edited this section to avoid mixing rejuvenation/aging with cell turnover during tissue homeostasis/renewal/restoration/regeneration (now within Box2, lines 788-840, with the renewal section specifically within lines 816-840).

Furthermore, multiple references missing some information (e.g. page numbers), and Kunh 1969 (line 34) is not in the reference list. But I believe these technicalities will be fixed during production.

We have added the missing Kuhn reference (now ref 1). Thank you.

Reviewer #3

In this manuscript Seifert, Duncan and Zayas share insights from a recent workshop that took place at the Banbury Center of Cold Spring Harbor, where big and open questions in the field of regenerative biology have been discussed. In an effort to raise these questions logically, the authors categorically described the key steps common to regenerative process and posed

different scenarios that could explain differences between regenerative and non-regenerative species when it comes to injury response and the outcome. Additionally, the authors highlighted the importance of increasing the diversity of model organisms used in regenerative research and enhancing cooperation between experts from different fields in an attempt to find answers.

First of all, I would like to thank the authors for putting this insightful piece together and allowing the missing members of the community to join in on the discussion. There is no doubt that such a discussion is very timely in light of all the new techniques that became available to the regenerative biology field recently.

Thank you.

Detailed comments:

Strengths

- 1-Excellent flow that keeps the reader engaged throughout.
- 2-The choice to focus on common questions related to regenerative biology rather than putting heavy emphases on a specific model organism or tissue.
- 3-High quality and depth of “known unknowns” discussed.
- 4-The content of Box 1-4 (very informative) and the experimental ideas in these sections.

Suggestions for improvement

5- While the quality and the context of the discussed questions are excellent, the obstacles limiting our progress in answering these questions are not crystallized. I would recommend the authors to clarify these obstacles more clearly. Are these obstacles merely due to technical limitations? Can we overcome these limitations using the modern technologies that the authors alluded to? As it is written it comes across as if it would be sufficient just to study a more diverse cohort of species. However, we are still lacking for example temporal resolution to capture mechanistic events happening in short time scale due to lack of imaging modalities that can longitudinally capture these events live in deep tissue over the course of many days that is required for certain examples of tissue regeneration.

This is a great point. We have expanded several sections (e.g., lines 48-54, 199-204, 265-268, 345-351, 411-426) to more clearly define what technical limitations exist and what questions require these tools.

6-In the abstract and introduction, the authors emphasize the importance of bringing together scientists and experts from various fields and describe the current workshop as a platform where this was achieved. However, the outcome is unclear to the reader. Could the authors please elaborate on key realizations that were made as a result of having technical experts together with developmental/evolutionary biologists?

We agree with the reviewer and have edited the text throughout the manuscript to expand on this suggestion (e.g., lines 48-54, 68-80, 199-204, 411-426, and the new Box 1 in lines 772-786).

7-One of the main take away messages is to increase the diversity of species used in research to enable comparisons that might answer some of the questions posed in the meeting. There are already excellent examples in the literature where researchers compared the regenerative process between different species (for example: PMID: 28632131, PMID: 24268695, PMID: 30462998). Could the authors elaborate on whether these comparisons have led to the expected outcome? And if not, why? Also, I assume that finding answers to the questions posed here requires coordinated action of many labs, perhaps in the form of a consortium, to ensure experiments are performed in a way where results are openly available and comparable (consistency in experimental design, technological platform used, bioinformatic analysis parameters etc). The authors partially touch upon this in Line 348-351. In this sense, this article would be an excellent starting point for discussion. Was there any suggestion in the meeting on how to handle this?

Thank you for raising these questions. We agree that in addition to highlighting the need to expand the breadth of quality work, we should highlight examples that support how comparative studies can lead to mechanistic insights. In light of this comment, we have updated the Concluding Remarks section to address this point and include a sentence specifically highlighting the examples referred by the referee as well as comparative studies from flatworms (see Lines 444-457 and citations 8 and 93-96)

8-Figures are not very self-explanatory. Please see more detailed comments below. Additionally, it might be helpful to have an extra column in Table 1 to clarify which steps belong to a) healing, b) blastema and c) morphogenesis.

We have updated Fig. 1 and Fig. 3 to make them more clear and added a new column to the table to denote the three categories of processes to which we refer in the text and Figs. 1 and 2.

Minor comments:

9- It might be wise for the authors to include a description of blastema in a Box. Do we consider formation of a blastema in internal organ regeneration such as the regeneration of the heart or spinal cord etc.?

We have edited the text when we bring up the blastema to address this point. However, the second clause here about internal organs is an excellent inquiry that is beyond the scope of our perspective. We think the question raised would be best dealt with in a review focused on internal organ regeneration.

10- The authors write that the conference attendees worked to formulate an ambitious plan to discover answers to new and enduring questions (Line 49-50). It is not clear what the plan is.

We have edited the text in response to this comment (e.g., the addition of Box 1, now found in lines 772-786; lines 48-54, and lines 411-426). Although we had multiple group discussions about embracing interdisciplinary collaborations, stating that we formulated a plan was misleading. This initial meeting led to generating questions. The next step is to organize a future

meeting to formulate plans to tackle these questions. However, we do provide ideas for testable hypotheses throughout the text.

11- Line 76: It would be nice to spell out what the new research organisms are for the readers who might not be as familiar.

We have added additional text and references in the Concluding Remarks section (VII). After discussion amongst ourselves, we felt it was difficult to offer particular organisms that should be pursued as this would be specific to individual investigators as it relates to their questions of interest.

12- Line 176: Newts are salamanders. Please revise to salamander limbs or detail further as salamanders, such as axolotl and newts or use urodele amphibians as in other parts of the manuscript.

We have corrected this (line 177). Thank you for pointing this out.

13- Line 181: The statement that “Collectively, these findings support the concept that the ability to accumulate cycling cells is a key feature that distinguishes wound healing processes from regeneration.” does not fit in where it currently is as the authors have just cited studies that showed that accumulation of cycling cells is not sufficient in the absence of certain signals.

We have edited this sentence in response to this comment (line 174-175).

14- Line 412-414: The authors write “As researchers, individually and collectively, we believe progress can be made toward understanding regenerative ability in the context of evolution by designing experiments to solve some of the problems presented above.” Could they expand on this through discussion of how new technologies and cross-disciplinary approaches might help in the design of these experiments? Were there any concrete suggestions within the group?

This is now included in the concluding remarks section (lines 411-426).

15- Line 423-424: could the authors please add references?

We now reference Seifert, A. W. & Voss, S. R. Revisiting the relationship between regenerative ability and aging. *BMC Biol* **11**, 2 (2013) in line 434 (ref 91).

16- Line 423-427: maybe should not be in concluding remarks but in the introduction?

We considered your suggestion to move the text. However, after extensively editing sections of the manuscript to incorporate your excellent suggestions, we decided the concluding remarks remained an appropriate place to discuss organism properties that should be considered when selecting other suitable models for comparative studies.

17- Line 817: Please write gene regulatory networks (GRNs).

Done (now line 921).

18- Figure 1: Panel B-E, please revise the Y axis label to “Process variation between regenerative species”.

This change has been made.

19- Figure 2: Panel C and D, please revise the Y axis label to “Process similarity between regenerative and non-regenerative species or species” or similar.

This change has been made.

20- Figure 3: Panel A, in the figure legends the authors write “specific genes are present and possibly activated during development across multiple species, but only induced after injury in regenerative species.” The cartoon does not reflect this statement. I suggest that the authors to add a sentence such as “potential developmental gene” to the figure. Additionally, in the figure the authors display differential epigenetic modifications that might be involved in gene regulation. It might be good to explain this in the figure legend.

We appreciate this feedback on how to make this figure clearer. We have removed the language about development from the figure legend, as this developmental process is not represented here. We have also added a Key explaining what the different ovals and modifications represent.

21- Figure 3: panel B, is gene Z supposed to be a regeneration specific gene?

No, in B, the enhancer is regeneration-specific but not gene Z. We have clarified this in the legend.

REVIEWERS' COMMENTS:

Reviewer #1 (Remarks to the Author):

The authors have responded well to the comments and revised the manuscript appropriately. It is an interesting and timely article. It is suitable that it be published as a "Perspectives" article instead of a Review article.

Reviewer #2 (Remarks to the Author):

The authors fully addressed my previous question. The manuscript is further improved following the comments from two other reviewers. I don't have further suggestions.

Reviewer #3 (Remarks to the Author):

I would like to thank the authors for the revisions they made. My concerns have been addressed and I find the manuscript greatly improved. I believe this will be a thought-provoking piece propelling discussions in the regenerative biology community.